# Fault-Tolerant Control Strategy for Sub-Modules Open-Circuit Fault of Modular Multilevel Converter

**Yaoxi Jiang** [1,2] , **Hongchun Shu** [2,*] **and Mengli Liao** [2]

1   The College of Information Engineering and Automation, Kunming University of Science and Technology, Kunming 650504, China
2   The State Key Laboratory Innovation Center for Smart Grid Fault Detection, Protection and Control Jointly, Kunming University of Science and Technology, Kunming 650504, China
*   Correspondence: kmshc@sina.com

**Abstract:** Modular multilevel converter (MMC) is a key device of high-voltage-direct circuit (HVDC) transmission system, the sub-module detection technology of which will directly influence the damage severity, and even the reliability of the whole system. In this paper, the open-circuit fault characteristics of an insulated gate bipolar transistor (IGBT) in a sub-module are analyzed, and a fault-tolerant optimal control strategy is proposed for the redundant hot-reserved MMC based on nearest-level modulation (NLM). A fault sub-module diagnosis and location strategy based on the deviation distance of the capacitor voltages is presented. After the faulty sub-module is removed, due to the asymmetric operation of the MMC, odd-order circulating currents are introduced in the faulty phase, in which the fundamental-frequency circulating current is the major component; the fundamental-frequency voltage related to the redundancy rate is injected into the faulty phase, which effectively suppresses the fundamental-frequency circulating current and harmonics in the faulty phase. The proposed method combines fault detection and fault ride-through steps, so it has the features of high reliability and high compatibility. Based on the Matlab/Simulink simulation model, the effectiveness of the proposed strategy is verified.

**Keywords:** modular multilevel converter (MMC); open-circuit fault; injection method; circulating current suppressing; fault-tolerant control

## 1. Introduction

Modular multilevel converter (MMC) is widely used in power transmission systems, offshore wind power grid connection and other fields attributing to its high modularity, and good output waveform quality [1–4]. However, a large number of cascaded sub-modules are potential failure points, which bring great challenges to the reliability of MMC [5]. Insulated gate bipolar transistor (IGBT) is the main element of MMC, and an IGBT fault is the most popular fault in MMC. The short-circuit fault of IGBT are much more serious, so the IGBT short-circuit detection circuit is becoming a standard component of MMC. An IGBT open-circuit fault has the characteristic of a long duration and therefore is difficult to monitor. Since there is no discharge circuit path, open-circuit faults can cause a significant increase in capacitor voltage in the faulty sub-module, which leads to cascading damage to the MMC system. In serious cases, the open-circuit fault sub-module may have severe over-voltage, which damages capacitors and healthy semiconductor switches, and even accompany an explosion, resulting in the system's shut down. Therefore, the rapid and timely detection and removal of faulty sub-module, shortening the fault detection time, and improving fault-tolerant operation control strategies are essential to ensuring the stable operation of MMC [6].

IGBT open-circuit fault diagnosis methods have hardware and software based methods [7]. Hardware based methods have faster diagnosis speed, but require more complex hardware circuits and control system [8–10]. Ref. [8] adds sub-modules set voltage sensors

to realize the fault detection. In [9,10], the extra sub-module voltage output sensor is added to detect the faulty sub-module. Although these methods are effective, they are not economical to implement due to the need to change the mature hardware structure of the half or full bridge sub-module.

Software-based methods are with no need for modifying sub-module structure or adding sensors. Most of these methods are mathematical model-based methods that estimate a specific current or voltage through different observers based on the MMC model, and then compare the estimated value with the measured value to determine if there is an open-circuit fault [11–14]. Literature [11] calculates the correlation coefficients which brings a heavy computation burden. Ref. [12] only considers and detects lower switch open-circuit fault in sub-module. The various switching states are investigated in [13], and a modified Pauta criterion is presented in ref. [14] to locate the faulty sub-module. Other software-based methods are artificial intelligence-based methods [7,15–17]. Ref. [7] proposes a sub-module switch open-circuit fault diagnosis method based on random forest algorithm. Support vector in [15], probabilistic neural network in [16], and 1-D convolutional neural networks in [17] are developed to realize sub-module open-circuit fault localization. Artificial intelligence-based methods require a large amount of data and calculation to train the neural networks, which accordingly result in low efficiency of fault detection.

After the faulty sub-module is detected and removed, the circulating currents due to the asymmetric operation of the arm are increased, and the ride-through control strategy should be put into place to improve the economy and reliability of MMC operation [18–22]. The circulating currents can be suppressed by controlling the energy balance of the arm [18], changing the displacement of the neutral point in the direction of the faulted phase voltage vector [19], introducing virtual resistors [20], etc. Reference [21] proposed a fault tolerance method based on zero-sequence voltage. Another fault tolerance method is proposed by injecting DC and AC components, which improves fault tolerance under multi-phase fault conditions [22].

Fault localization methods proposed above are based on the fact that the capacitor voltage of the faulty sub-module can become much higher than that of a normal sub-module based on pulse width modulation (PWM). The modulation strategy for MMC with a large number of sub-modules is widely used is nearest-level modulation (NLM), because of its advantages of low switching frequency and easy implementation. For NLM-based MMC with a large number of sub-modules, the capacitor voltage of the faulty sub-module is not too different from that of the normal sub-module as PWM-based MMC. There are few researches for NLM-based MMC in the existing literature [23,24]. Therefore, NLM-based MMC is studied in this paper, and presents a diagnosis and fault ride-through control method for sub-module open-circuit fault.

The proposed method in this paper combines fault localization and fault ride-through steps, so it has the features of high reliability, high compatibility, and high effectiveness. In [23], the diagnosis method based on NLM requires an additional inductor. As soon as the faulty sub-module is removed, a sub-module in the other arm is removed to achieve the goal of the energy balance among the arms, and the fault-detecting period is designed as 40 milliseconds. This paper utilizes the measured capacitor voltages to locate faults before the fault sub-module capacitor voltage is charged very high. Single sub-module faults are the most common, so only single sub-module faults are considered in this paper. This method just requires little calculation cost, because the capacitor voltage deviation distance based on the capacitor voltages is calculated no more than $n$ times to detect a fault sub-module in an arm with $n$ sub-modules.

In order to realize fault ride-through control, as the faulty sub-module is detected and removed in ref. [24], a sub-module in the other arm is also removed to achieve the energy balance among arms; arm current is reshaped in [25] to suppress sub-module over-voltage just before the fault sub-module is detected. In order to improve the economy of the MMC as much as possible, this paper adopts the strategy of only removing the faulty sub-module

which brings about the unbalance operation of MMC, then the fundamental-frequency voltage related to the redundancy ratio is injected into the faulty phase to compensate the unbalanced circulating voltage.

This paper is organized as follows: Section 2 gives the principle of the NLM-based hot-reserved MMC. Section 3 studies IGBT open-circuit fault for the NLM-based hot-reserved MMC. IGBT open-circuit fault detection method and tolerance control strategy are proposed in Section 4. The effectiveness of the proposed strategy is verified in Section 5. Finally, a conclusion is presented in Section 6.

## 2. The Principle of Hot-Reserved MMC

### 2.1. The Topology of MMC

The topology of the half-bridge MMC is shown in Figure 1. In this paper, the lower subscript $n$ and $p$ represent the variables of the lower and upper arm separately. Each arm is formed by $n$ half-bridge sub-modules (HBSMs) and reactance in series. $R_0$ and $L_0$ are the equivalent resistance and inductance of the arm inductor; $R_s$ and $L_s$ are the equivalent resistance and inductance of the AC side; $i_{pa}$ and $i_{na}$ are the arm currents of phase-a; $u_{pa}$ and $u_{na}$ are the arm voltages of phase-a. From the equivalent circuit of a-phase of MMC shown in Figure 2, there are:

$$u_{\Sigma} = u_{na} + u_{pa} \tag{1}$$

$$u_{\Sigma} = U_{dc} - 2L_0 \frac{di_{cira}}{dt} - 2R_0 i_{cira} + \frac{1}{C_{ph}} \int i_{cira} dt \tag{2}$$

where $i_{cira}$ and $u_{cira}$ are the circulating current and voltage of phase-a, $u_{\Sigma}$ is the common mode component of phase voltage, and $C_{ph}$ is the phase equivalent capacitance. According to the principle of energy storage and the total withstanding voltage unchanged, the capacitors of all $n$ sub-modules of the entire phase-a are equivalent to a capacitor $C_{ph}$:

$$\frac{1}{2} C_{ph} U_{dc}^2 = 2n \frac{1}{2} C U_c^2 \tag{3}$$

$$C_{ph} = \frac{2C}{n} \tag{4}$$

where $C$ is capacitance of the sub-module capacitor, and $U_c$ is the rated voltage of the sub-module capacitor. Ignoring the bridge arm inductance:

$$u_{cira} = u_{\Sigma} - U_{dc} = \frac{1}{C_{ph}} \int i_{cira} dt \tag{5}$$

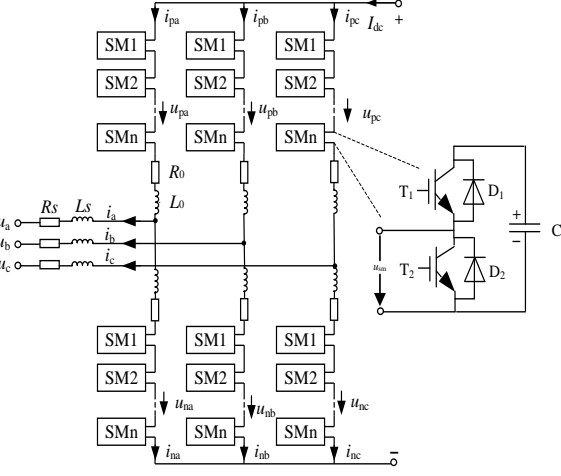

**Figure 1.** Topology of MMC.

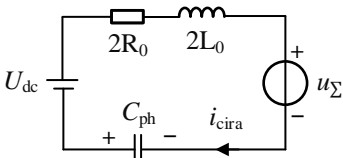

**Figure 2.** Equivalent circuit of a phase of MMC.

Equation (5) shows the relationship between the common mode component $u_\Sigma$ of the phase voltage, the circulating voltage $u_{cira}$ and the circulating current $i_{cira}$. The fluctuation of $u_\Sigma$ brings about the fluctuations of $u_{cira}$ and $i_{cira}$, so the way to suppress the circulating current $i_{cira}$ is the same to suppress the fluctuation of $u_\Sigma$.

*2.2. Mathematical Model of Hot-Reserved MMC*

The redundant sub-modules are utilized to realize the cold reserved or hot-reserved modes to improve the reliability of MMC. The redundant sub-module hot-reserved mode is often used to improve the fault tolerance performance of MMC. The hot-reserved mode puts all sub-modules including the redundant sub-modules into operation. When an open circuit fault occurs, the faulty sub-module is bypassed, and the number of inserting sub-modules is not reduced to ensure the uninterrupted operation of the MMC, which has the advantages of a simple, reliable, and seamless fault-ride through process. However, because the faulty sub-modules were bypassed, the number of sub-modules in the arms is different. Asymmetric operation of the MMC brings about the increasing circulating currents, distorting output currents, and larger ripples in capacitor voltages, etc.

There are $(n + k)$ sub-modules in each bridge arm of the hot-reserved MMC, in which $n$ sub-modules are necessary to maintain the output level of the bridge arm, and other $k$ sub-modules are hot-reserved sub-modules. Therefore, from the perspective of the DC side voltage, at least $k$ sub-modules in each bridge arm are in an idle state at any time. Redundancy $r$ refers to the proportion between the number of redundant sub-modules and the number of sub-modules necessary to maintain the normal operation of the system:

$$r = \frac{k}{n} \times 100\% \tag{6}$$

when the faulty sub-modules are bypassed, the redundancy $r_f$ of the faulty bridge arm is:

$$r_f = \frac{k - n_f}{n} \times 100\% \tag{7}$$

where $n_f$ is the number of faulty sub-modules. For ensuring that the number of output levels of the faulty arm voltage remains unchanged, the sub-module fault redundancy rate $r_f$ needs to be greater than 0. $r_f$ is 0 means that the arm has no redundant sub-modules.

Taking phase-a as an example shown in Figure 1, the relationships among the alternating current $i_a$, the circulating current $i_{cira}$ and the upper and lower arms currents $i_{pa}$ and $i_{na}$ are:

$$\begin{cases} i_a = i_{pa} - i_{na} \\ i_{cira} = 0.5(i_{pa} + i_{na}) \end{cases} \tag{8}$$

Assuming that the voltage modulation ratio of MMC is 1, the AC voltage and current expressions are:

$$\begin{cases} u_a = \frac{1}{2}U_{dc}\sin(\omega t) \\ i_a = I_m\sin(\omega t - \theta) \end{cases} \tag{9}$$

where $I_m$ is the amplitude of the alternating current; $\omega$ is the angular frequency; $\theta$ is the angle between the output current and voltage. It can be obtained from (1), (2), (5), and (8):

$$\begin{cases} \frac{1}{2}(U_{dc} - u_{pa} - u_{na}) = L_0\frac{di_{cira}}{dt} + R_0 i_{cira} \\ \frac{1}{2}(u_{na} - u_{pa}) = \frac{1}{2}L_0\frac{di_a}{dt} + \frac{1}{2}R_0 i_a + L_s\frac{di_a}{dt} + R_s i_a \end{cases} \tag{10}$$

where $R_0$ is the equivalent resistance of the bridge arm. The voltage components on the left side of Equation (10) determine the circulating current and the output current, respectively. By controlling these two voltage components, the control of the MMC output current and circulating current can be achieved. Define phase-a circulating voltage $u_{cir\_a}$:

$$u_{cir\_a} = \frac{1}{2}(U_{dc} - u_{pa} - u_{na}) \tag{11}$$

The upper and lower arm currents are:

$$\begin{cases} i_{pa} = \frac{1}{2}i_a + i_{cira} = \frac{1}{2}i_a + I_{da} + \sum_{h=1}^{\infty} i_h \\ i_{na} = -\frac{1}{2}i_a + i_{cira} = -\frac{1}{2}i_a + I_{da} + \sum_{h=1}^{\infty} i_h \end{cases} \tag{12}$$

where $I_{da}$ is the DC component in the circulating current, and $i_h$ is the $h$th harmonic component of the circulating currents.

For the hot-reserved MMC, the number of sub-modules put into in the upper and lower arms are $(n + k - n_{fp})$ and $(n + k - n_{fn})$ respectively, where $n_{fp}$ and $n_{fn}$ represent the number of faulty sub-modules that have been bypassed in the upper and lower arms. For the nearest level modulation (NLM) based MMC, the average switching function of sub-module in the upper and lower arms are:

$$\begin{cases} S_{pa} = \frac{1}{2}\frac{n}{n+k-n_{fp}}(1 - \sin(\omega t)) \\ S_{na} = \frac{1}{2}\frac{n}{n+k-n_{fn}}(1 + \sin(\omega t)) \end{cases} \tag{13}$$

According to (12) and (13), the average sub-module capacitors currents $i_{cpa}$ and $i_{cna}$ are:

$$\begin{cases} i_{cpa} = \frac{1}{2}\frac{n}{n+k-n_{fp}}(1 - \sin(\omega t)) \cdot i_{pa} \\ i_{cna} = \frac{1}{2}\frac{n}{n+k-n_{fn}}(1 + \sin(\omega t)) \cdot i_{na} \end{cases} \tag{14}$$

Then the fluctuation of the capacitor voltage of the upper and lower arm sub-modules can be expressed as:

$$\begin{cases} \Delta u_{cpa} = \frac{1}{C}\int i_{pa}dt \\ \Delta u_{cna} = \frac{1}{C}\int i_{na}dt \end{cases} \tag{15}$$

Considering that the average voltage of the sub-module capacitor has been maintained at the rated voltage $U_{dc}/n$, the output voltages $u_{smpa}$ and $u_{smna}$ of the sub-module are:

$$\begin{cases} u_{smpa} = \frac{1}{2}\frac{n}{n+k-n_{fp}}(1 - \sin(\omega t)) \cdot (\frac{U_{dc}}{n} + \Delta u_{cpa}) \\ u_{smna} = \frac{1}{2}\frac{n}{n+k-n_{fn}}(1 + \sin(\omega t)) \cdot (\frac{U_{dc}}{n} + \Delta u_{cna}) \end{cases} \tag{16}$$

Ignoring the arm inductance voltage, the output voltages of bridge arms are:

$$\begin{cases} u_{pa} = (n + k - n_{fp})u_{smpa} \\ u_{na} = (n + k - n_{fn})u_{smna} \end{cases} \tag{17}$$

The circulating voltage $u_{cir\_a}$ of phase-a can be deduced:

$$u_{cir\_a} = u_{cir\_sym} + u_{cir\_asym} \tag{18}$$

$$u_{\text{cir\_sym}} = \frac{n}{8C}\left(\frac{1}{1+r_{\text{fp}}} + \frac{1}{1+r_{\text{fn}}}\right) \times$$
$$\left\{\sin(\omega t)\int\left[\frac{2I_{\text{da}}\sin(\omega t-\theta)}{\cos(\theta)} - I_{\text{da}}\sin(\omega t) - \sin(\omega t)\sum_{h=1}^{\infty}i_h\right]dt - \int\left[\sum_{h=1}^{\infty}i_h + \frac{2I_{\text{da}}\cos(2\omega t-\theta)}{\cos(\theta)}\right]dt\right\} \quad (19)$$

$$u_{\text{cir\_asym}} = \frac{n}{8C}\left(\frac{1}{1+r_{\text{fp}}} - \frac{1}{1+r_{\text{fn}}}\right) \times$$
$$\left\{\sin(\omega t)\int\left[\sum_{h=1}^{\infty}i_h + \frac{2I_{\text{da}}\cos(2\omega t-\theta)}{\cos(\theta)}\right]dt - \int\left[\frac{2I_{\text{da}}\sin(\omega t-\theta)}{\cos(\theta)} - I_{\text{da}}\sin(\omega t) - \sin(\omega t)\sum_{h=1}^{\infty}i_h\right]dt\right\} \quad (20)$$

where $r_{\text{fp}}$ and $r_{\text{fn}}$ represent the redundancy ratios of the upper and lower arms in a phase. It can be seen that the circulating voltage is not only related to the circuit parameters, but also related to the redundancy of the arms. The circulating voltage can be divided into two parts: the symmetric circulating voltage component $u_{\text{cir\_sym}}$ and the asymmetric circulating voltage component $u_{\text{cir\_asym}}$. When the number of sub-modules of the arms are equal, that is, when the redundancy of the upper and lower arms is equal, the phase operates symmetrically. The asymmetric circulating voltage $u_{\text{diff\_asym}}$ is 0, and the symmetrical circulating voltage $u_{\text{cir\_sym}}$ does not contain odd-order harmonic circulating currents, so the harmonic circulating voltage contains only 2, 4, 6 and other even-order harmonic voltage components.

However, if the number of sub-modules in the upper and lower arms in a phase are different, that is, if the redundancy of the arms is not equal, the phase operates asymmetrically, and the two parts of the circulating voltage are not zero. Since the redundancies of the arms is not much different, the symmetrical circulating current voltage $u_{\text{cir\_sym}}$ is still the main component, containing even-order harmonic components the same as the symmetrical operation. Asymmetric circulating voltage $u_{\text{cir\_asym}}$ contains fundamental component, and other odd-order components of $h + 1$ and $h - 1$ harmonic voltages, which are generated by the fundamental component multiplied by $h$th circulating voltages.

The conventional circulating current controller can only suppress the second circulating current and has no effect on the odd-order circulating current, which causes the DC side current to fluctuate and increase the power loss of MMC. Therefore, after the fault sub-module is removed, the faulty phases operate asymmetrically, and the odd-order circulating voltages needs to be suppressed.

## 3. Analysis of Sub-Module IGBT Open Circuit Fault

### 3.1. $T_1$ Open-Circuit Fault

Thermo-mechanical induced stresses and significant junction temperature fluctuations are the main causes of open-circuit failure in IGBT. Other common causes of open-circuit failure include gate driver failure, electrical disconnection, etc. The sub-module IGBT open-circuit faults have types of $T_1$ and $T_2$ open-circuit fault, respectively, as shown in Figure 3. The relationship between the sub-module output voltage $u_{\text{sm}}$, the charge and discharge status of the capacitor, the bridge arm current $i_{\text{arm}}$, and the phase common mode volt age $u_{\sum}$ is shown in Table 1.

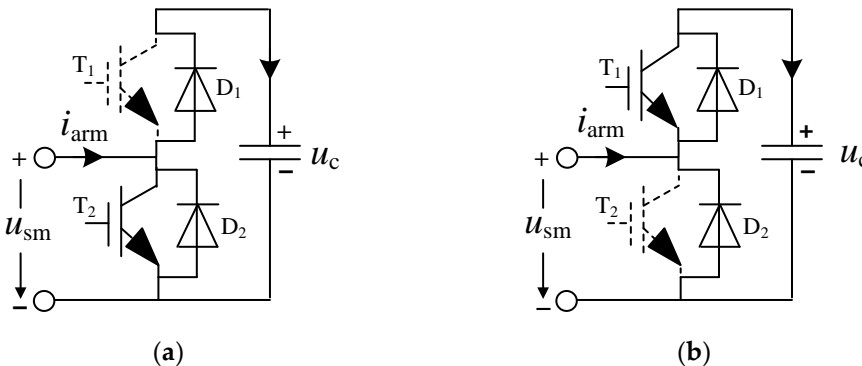

**Figure 3.** Half-bridge sub-module IGBT open circuit fault. (**a**) $T_1$ open-circuit fault; (**b**) $T_2$ open-circuit fault.

**Table 1.** IGBT Open Circuit Fault Characteristics of Half-bridge Sub-module.

| Fault Types | $i_{arm}$ | Switching Signal | $u_{sm}$ | Capacitor State (Fault) | Capacitor State (No Fault) | $u_\Sigma$ |
|---|---|---|---|---|---|---|
| $T_1$ fault | >0 | 1 | $u_c$ | charge | charge | $N \times u_c$ |
| | | 0 | 0 | bypass | bypass | $N \times u_c$ |
| | <0 | 1 | 0 | bypass | discharge | $(N-1) \times u_c$ |
| | | 0 | 0 | bypass | bypass | $N \times u_c$ |
| $T_2$ fault | >0 | 1 | $u_c$ | charge | charge | $N \times u_c$ |
| | | 0 | $u_c$ | charge | bypass | $(N+1) \times u_c$ |
| | <0 | 1 | $u_c$ | discharge | discharge | $N \times u_c$ |
| | | 0 | 0 | bypass | bypass | $N \times u_c$ |

For the NLM-based MMC, since $T_1$ is always in the off state when the sub-module outputting 0, the $T_1$ open-circuit fault only affects the state of the sub-module outputting $u_c$ as showed in Figure 3a. When the sub-module outputs $u_c$, it depends on the direction of the bridge arm current $i_{arm}$. When the current $i_{arm}$ is negative, if the $T_1$ of $i$-th sub-module is open, the current will automatically conduct through $D_2$ and the faulty sub-module outputs voltage 0; when the bridge arm current is positive, the current passes through the diode $D_1$, and the open circuit of $T_1$ does not affect the output of the sub-module, and the output voltage of the sub-module is equal to its instantaneous capacitance voltage $u_c$ at this time.

Once the voltage of the faulty sub-module is higher than the capacitor voltage of other sub-modules, according to the nearest level modulation (NLM) algorithm, in order to equalize the voltage of all the capacitors, when the arm current is positive, the faulty sub-module is bypassed and the capacitor voltage is maintained; when the arm current is negative, the capacitor of the fault sub-module is always inserted to discharge according to NLM; however, the $T_1$ open-circuit fault makes the fault capacitor bypassed, so the fault capacitor voltage is maintained. Accordingly, due to the open-circuit of $T_1$, the capacitor of the faulty sub-module can be charged, but cannot be discharged, resulting in a significant increase in its voltage.

Therefore, for the NLM-based MMC, the capacitor voltage of the sub-module with the $T_1$ fault is not always charged and increased, but keeps charging to maintain the largest one in the arm.

Since the faulty sub-module cannot be inserted normally, when the arm current is negative, although the rest of the sub-modules are normally switched on and off, the arm voltage does not include the faulty sub-module capacitor voltage, that is, the arm voltage is reduced by $u_c$, the fault phase common mode voltage $u_\Sigma$ is $(N-1) \times u_c$.

### 3.2. $T_2$ Open-Circuit Fault

The $T_2$ open-circuit fault only affects the state that the sub-module outputs 0, in which $T_2$ is on. When the $T_2$ open circuit fault occurs in the sub-module outputting 0, the positive arm current flows into the faulty sub-module through the diode $D_1$ and the faulty sub-module outputs $u_c$ as shown in Figure 3b. The capacitor of the faulty sub-module changes from the bypass state to the charged state, so the capacitor voltage increases significantly. Since the faulty sub-module cannot be bypassed when the arm current is positive, the arm voltage includes the output voltage of the faulty sub-module, that is, the arm voltage of the phase increases $u_c$, so the common mode voltage $u_\sum$ is $(N + 1) \times u_c$.

When the bridge arm current is negative, the current passes through $D_2$. The capacitor is bypassed and the fault sub-module still outputs 0, so the open-circuit of $T_2$ does not affect the state of the sub-module capacitor at this time.

Once the voltage of the faulty sub-module is higher than the other sub-module capacitor voltages, when the bridge arm current is positive, the faulty sub-module is bypassed, but the capacitor will be charged at this time; when the arm current is negative, the capacitor is normally put into operation in discharged or bypassed state according to NLM. The voltage variance $\Delta u_c$ of the faulty sub-module capacitor in one arm current cycle is expressed as:

$$\Delta u_c = \frac{1}{C} \int_{i_{arm}>0} i_{arm} dt + \frac{1}{C} \int_{i_{arm}<0} i_{arm} dt \qquad (21)$$

Figure 4 shows the arm current of the inverter MMC. The direct current component $I_{dca}$ makes the positive current time range larger than the negative current time range, and the positive current has a larger current level than the negative current.

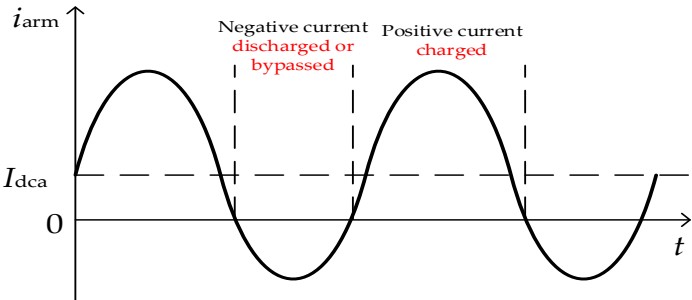

**Figure 4.** The arm current of inverter MMC.

The capacitor of the $T_2$ open-circuit faulty sub-module is always in a charging state when the bridge arm current is positive, and it is discharged or bypassed according to NLM when the bridge arm current is negative. The inverter MMC has a larger time range and a larger current level in the positive bridge arm current, and there is not enough time to discharge it in the negative bridge arm current to offset the voltage increase, so the faulty sub-module capacitor voltage will continue to increase, and it needs to be removed in time.

To sum up, whether it is a $T_1$ or $T_2$ open-circuit fault, the capacitor voltage of the faulty sub-module has tendency to increase and is the largest in the faulty arm, which may lead to output distortion and the failure of the MMC. For PWM-based MMC, the increasing tendency of faulty capacitor voltage is much more distinct, and its fault localization methods are not feasible for NLM-based MMC [23,24], so an effective diagnostic scheme for NLM-based MMC has to be proposed.

Furthermore, it can be seen from Table 1 that both $T_1$ fault and $T_2$ fault cause the phase common mode voltage $u_\sum$ to fluctuate, and the faulty phase circulating voltage and current fluctuations also increase according to (5).

## 4. Sub-Module Fault Tolerance Control Strategy

### 4.1. Diagnosis of Single Sub-Module Open-Circuit Fault

Due to NLM and the capacitor voltage equalization algorithm, the periodic variation trends of all capacitor voltages of all sub-modules of each arm are the same. When the MMC just transfers reactive power, the capacitor voltage fluctuation of the sub-module is the largest [26]:

$$max|\Delta u_{\mathrm{c}}| = \frac{1}{3}\frac{S_{\mathrm{VN}}}{N\omega C_{\mathrm{sm}}U_{\mathrm{sm}}} \tag{22}$$

where $S_{\mathrm{vN}}$ is the rated capacity of the MMC and $N$ is the number of sub-modules in an arm. $U_{\mathrm{sm}}$ is the rated sub-module capacitor voltage and equals to $U_{\mathrm{dc}}/N$. $\Delta u_{\mathrm{c}}$ is the magnitude of the voltage fluctuation of the sub-module capacitor.

When an open-circuit fault occurs in a sub-module, the capacitor voltage of the faulty sub-module deviates from the capacitor voltages of other normal sub-modules. Therefore, the deviation distance $d(u_{ci}, u_{cj})$ of the capacitor voltages of the $i$th and $j$th sub-modules in an arm is defined as:

$$d(u_{ci}, u_{cj}) = \sum_{l=1}^{m} |u_{ci}(l) - u_{cj}(l)| \tag{23}$$

$u_{ci}(l)$ and $u_{cj}(l)$ ($i, j = 1, 2, 3 \ldots n + k$) is the capacitor voltage value of the $i, j$th sub-module of the arm, and $m$ is the number of capacitor voltage sampling value in the window $\Delta T$. It can be seen that $d(u_{ci}, u_{cj})$ represents the accumulation of the deviation of the capacitor voltage of the two sub-modules in the time window $\Delta T$. The deviation distance between the normal sub-modules is small, and it is larger between the faulty sub-module and normal sub-modules. When the deviation distance exceeds a predefined threshold $d_{\mathrm{th}}$, the MMC has a sub-module fault. A higher value of $d_{\mathrm{th}}$ decreases the probability of false diagnosis brought by transient disturbances, but results in longer detection time windows $\Delta T$.

Since the probability of multiple sub-modules open-circuit faults is much smaller than the probability of a single sub-module fault, this paper considers the situation of only one sub-module fault, for which the flowchart is presented in Figure 5. The calculation of the capacitor voltage deviation distance is based on the capacitor voltage of the first sub-module, and the deviation distance $d(u_{ci}, u_{cj})$ is calculated from the capacitor voltage of the other sub-modules respectively. If the first sub-module SM$_1$ fails, all $(n - 1)$ deviation distances are more than the threshold $d_{\mathrm{th}}$. If only one deviation distance $d(u_{c1}, u_{ci})$ is more than the threshold, SM$_i$ is the faulty sub-module. The open-circuit fault

Choosing an appropriate threshold $d_{\mathrm{th}}$ is the key to achieve fast detection and avoid misdiagnosis. Most of literature use empirical thresholds. This paper sets:

$$d_{\mathrm{th}} = a \times m \times max|\Delta u_{\mathrm{c}}| \tag{24}$$

where $a$ is the correction coefficient, which can avoid wrong execution of fault detection procedure. In this paper, the value of a is 1.2. $m$ is the number of capacitor voltage sampling in the window $\Delta T$. The largest capacitor voltage fluctuation $max|\Delta u_{\mathrm{c}}|$ is calculated by (22).

The fault location time is the time window $\Delta T$. If $\Delta T$ is too short, the characteristic of the deviation distance $d(u_{c1}, u_{ci})$ is not obvious. If $\Delta T$ is too long, the fault location time is long, and the over-voltage of the faulty sub-module capacitor is serious. If there are no additional sensors, faulty sub-module detection time generally is tens of milliseconds. If there are additional sensors introduced, the fault detection time is also reduced. However, additional sensors mean higher hardware cost and more complex control system. Therefore, a trade-off between faster speed and lower cost is considered in this paper. Mathematical model-based methods require around 40–100 ms [11–14], and hardware-based methods require around 40 ms [8–10,25]. The fault sub-module methods in the literature [23,24] are also based on NLM, and the fault detection times are set at 40 ms in [23] and 20 ms in [24], separately. In this paper, considering both the anti-misdiagnosis and the reduction of detection time, the time window $\Delta T$ is set to 20ms, the detection speed is fast, no

misdiagnosis occurs, and the detection method has the merit of simplicity due to the low data-volume feature. of a sub-module can be diagnosed in real time using the sliding window method.

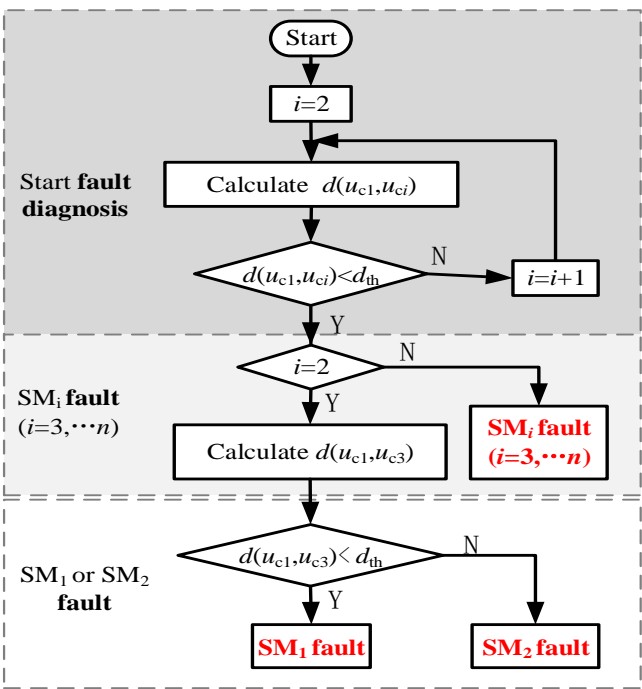

**Figure 5.** The flow chart of single sub-module fault diagnosis.

Since the sub-module capacitor voltage value used already exists in the MMC control system, no additional hardware resources are required. The amount of calculation of deviation distance is less than that of machine learning, and even smaller than that of correlation coefficient. The calculation of the capacitor voltage deviation distance in (24) is calculated $n$ times at most for an arm with $n$ sub-modules. In addition, because the fault diagnosis method is based on data analysis and is not dependent on the MMC mathematical model, it is not sensitive to the uncertainty of system parameters and has strong robustness.

The flowchart of single sub-module fault diagnosis in Figure 5 is just applicable to the single sub-module fault in an arm. The detecting method for multiple sub-modules faults is much more complex, and has a heavy computation burden. Single sub-module faults are the most common, so only single sub-module faults are studied and simulated in this paper.

### 4.2. Removal Strategy of Faulty Sub-Modules

When an open circuit fault occurs in MMC, there are two removal strategies for hot-reserved mode: the symmetrical bypass method and the asymmetric bypass method. All sub-modules are put into normal operation according to NLM. The symmetrical bypass method simultaneously removes the same number of sub-modules in each arm, so six arms still have the same number of sub-modules in operation. This method does not introduce new odd-order circulating currents. As long as the number of faulty sub-modules does not exceed the number of redundant sub-modules, the MMC operates normally, but the utilization rate of the sub-modules is reduced, which is uneconomical.

However, the asymmetric bypass method only removes the faulty sub-module, which can maximize the utilization of sub-modules. At the same time, the asymmetric bypass method causes asymmetric operation among arms resulting in new odd-order circulating currents. Therefore, new odd-orders circulating currents especially the fundamental-frequency circulating current should be suppressed to reduce the power loss of MMC.

### 4.3. Fundamental-Frequency Circulating Current Suppression Strategy

When an open-circuit sub-module fault occurs in a hot-reserved MMC, this paper adopts the asymmetric removal strategy that only bypasses the faulty sub-module. After the faulty sub-module is removed, the number of sub-modules that can be put into the faulty arm is reduced, the structure of the MMC is no longer symmetrical, the asymmetric circulating current voltage $u_{\text{cir\_asym}}$ is no longer 0, and the circulating current voltage $u_{\text{cir\_a}}$ has odd-order circulating voltages in which the fundamental-frequency component is at its maximum and needs to be suppressed. A fault-tolerant control method of injecting the fundamental-frequency voltage into the faulty phase is presented in this paper.

The injection method was used to suppress the second-order circulating current for symmetrically operating MMC [20,21]. According to the previous analysis of Equations (18)–(20), in the case of sub-module open-circuit failure and ignoring the harmonics, the symmetrical circulating current component has only DC and even-order frequency components, and the asymmetrical circulating current has only odd-order frequency components.

$$u_{\text{cir\_a}} = u_{\text{cir\_even}} + u_{\text{cir\_odd}} \tag{25}$$

$$u_{\text{cir\_even}} = \frac{n}{8C}\left(\frac{1}{1+r_{\text{fp}}} + \frac{1}{1+r_{\text{fn}}}\right) \times \left[\frac{2I_{\text{da}}\sin(2\omega t-\theta)}{\omega\cos(\theta)} - \frac{I_{\text{da}}\tan(\theta)}{\omega} - \frac{I_{\text{da}}\sin(2\omega t)}{2\omega}\right] \tag{26}$$

$$\begin{aligned}u_{\text{cir\_odd}} = {}& \frac{n}{8C}\left(\frac{1}{1+r_{\text{fp}}} - \frac{1}{1+r_{\text{fn}}}\right) \times \\ & \left[-\frac{I_{\text{da}}}{2\omega}\cos(3\omega t-\theta) + \frac{5I_{\text{da}}}{2\omega\cos(\theta)}\cos(\omega t-\theta) - \frac{I_{\text{da}}}{\omega}\cos(\omega t)\right]\end{aligned} \tag{27}$$

The circulating voltage includes the even-order component $u_{\text{cir\_even}}$ and the odd-order component $u_{\text{cir\_odd}}$; the even-order component mainly includes the DC component and the second-order frequency component; the odd-order component mainly includes the fundamental-frequency and the third-order frequency component. The fundamental-frequency circulating component in Equation (27) is injected in the fault phase to suppress the circulating current voltage of the faulty phase, so as to realize the fault-tolerant control of the open-circuit fault of the sub-module. The fault-tolerant control flowchart is illustrated in Figure 6.

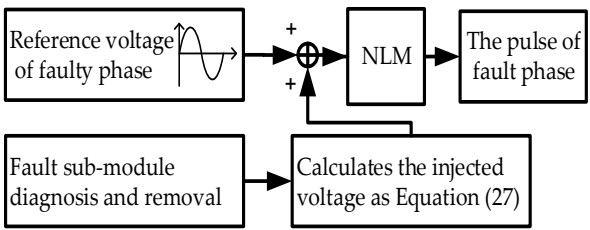

**Figure 6.** The flowchart of suppressing the fundamental circulating current.

The flowchart of suppressing the fundamental circulating current in Figure 6 is applicable to single and multiple sub-modules faults. After the fault sub-modules are removed, the redundancy rates are different for single and multiple sub-modules faults, and thus the amplitudes of the injected fundamental frequency in Equation (27) are different for fault ride-through control.

## 5. Simulation Verification

To verify the performance for the proposed sub-module faults diagnosis and fault-tolerant control strategy, a simulation study is carried out on a 21-levels inverter. The simulation parameters are given in Table 2. The modulation method is nearest-level modulation (NLM) based on a capacitor voltage sorting equalization strategy, and each arm has 2 redundant sub-modules.



**Table 2.** Simulation Parameters of MMC System.

| Item | Symbol | Value |
|---|---|---|
| System rated capacity/VA | $S$ | 200 |
| AC voltage peak/kV | $u_j$ ($j$ = a, b, c) | 200 |
| DC voltage/kV | $U_{dc}$ | $\pm 200$ |
| Arm inductance/mH | $L_0$ | 10 |
| The number of sub-modules in an arm | $N$ | 20 + 2 |
| Sub-module capacitance/mF | $C$ | 10 |

This paper only considers the situation of a single sub-module fault. It is set that a sub-module of the upper arm of phase-a fails. After an open-circuit fault occurs at 2 s, the circulating current fluctuation and distortion rate of the faulty phase increase sharply. The capacitor voltage sampling data collected in 20 ms after the fault occurs are used to calculate the capacitor voltage deviation distance. When the faulty sub-module is located and removed at 2.02 s, the remaining 21 sub-modules are put into balanced operation. The fundamental-frequency voltage is injected to realize fault-tolerant control at 2.5 s.

Figure 7 shows the $T_1$ fault simulation waveforms. Before the fault occurs at 2 s, the circulating current is mainly the second-order harmonic component as Figure 7a shows; when the sub-module fails at 2 s, the circulating current fluctuation increases; when the faulty sub-module is detected and removed at 2.02 s, the circulating current fluctuation decreases, and the fundamental-frequency circulating current is obviously introduced; when the fundamental-frequency voltage injection method is put into control at 2.5 s, the circulating current fluctuation is effectively suppressed.

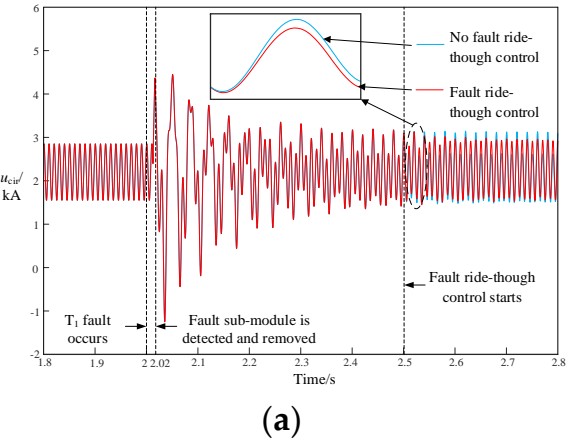

**(a)**

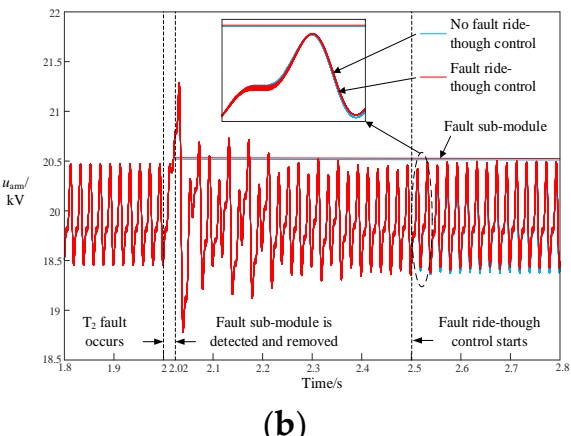

**(b)**

**Figure 7.** Simulation waveforms for $T_1$ open-circuit fault (**a**) Circulating current of faulty phase; (**b**) Capacitor voltages of faulty arm.

Figure 7b represents the capacitor voltages of the faulty arm. It can be observed that when a $T_1$ open-circuit fault occurs at 2 s, the capacitor voltage fluctuations of all sub-moules increase. After the faulty sub-module is detected and removed at 2.02 s, the capacitors of the other 21 sub-modules are balanced and put into operation according to NLM. After the fundamental-frequency voltage is injected into the fault phase at 2.5 s, the newly introduced fundamental-frequency harmonic current is suppressed, and the voltages of all 21 capacitors are shifted upward as a whole. Because the total energy stored in the arm remains unchanged, the energy stored in a single capacitor increases.

Figure 8 shows the circulating current and the faulty arm waveforms of the $T_2$ open-circuit fault. It can be seen that, similar to the $T_1$ fault, after the faulty sub-module is detected and removed, the fundamental frequency injection fault-tolerant control strategy can suppress the asymmetric circulating current without affecting the voltage balance of the fault arm capacitor voltage.

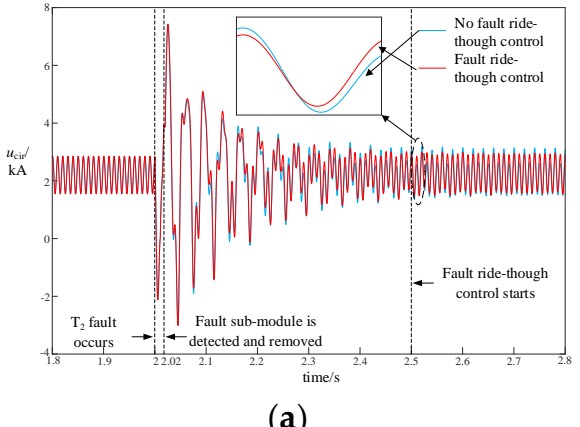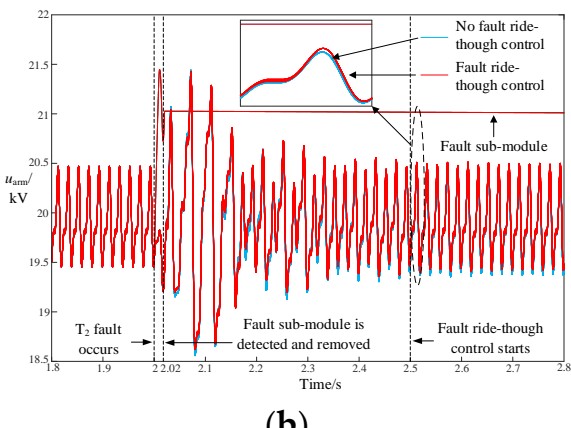

**(a)**  **(b)**

**Figure 8.** Simulation waveforms for T$_2$ open-circuit fault. (**a**) Circulating current of faulty phase; (**b**) Capacitor voltages of faulty arm.

Figure 9 shows the Fourier analysis of circulating currents, in which the diagrams show no faults in Figure 9a, no fault-tolerant control in Figure 9b, and the proposed fault-tolerant control in Figure 9c, respectively. It can be seen that there are mainly DC and second-order frequency components in the circulating currents before the fault occurs; after the fault sub-module is removed, there is a fundamental-frequency component in the circulating current with an amplitude of 261.6 A; when the proposed fundamental-frequency voltage injection fault-tolerant control is put in, the amplitude of the fundamental-frequency component is reduced to 26.9 A, and the total harmonic distortion (THD) is also decreased from1.55% to 1.36%.

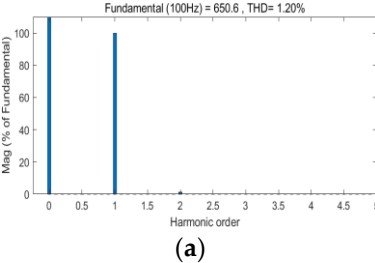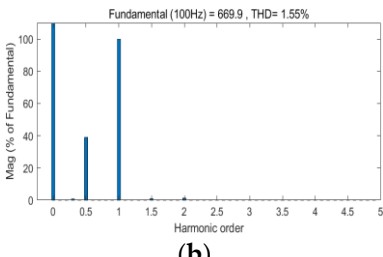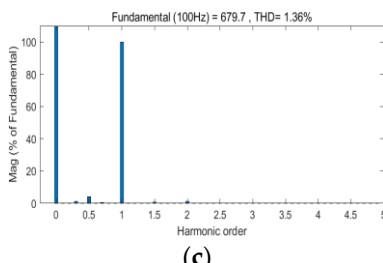

**(a)**  **(b)**  **(c)**

**Figure 9.** FFT analysis diagram of the circulating currents. (**a**) No fault occurs; (**b**) No fault-tolerant control after the fault sub-module is removed; (**c**) Proposed fault-tolerant control after the fault sub-module is removed.

## 6. Conclusions

This paper proposes an IGBT open-circuit fault diagnosis method by using the deviation distance of the capacitor voltages of the sub-modules, and a circulating current suppression method of injecting the fundamental-frequency voltage into the faulty phase. This method has two steps. The first step is the detection of the fault sub-module, and the second step is the fault-tolerant control. The proposed method in this paper combines fault localization and fault ride-through steps, so it has the advantages of high reliability and high compatibility.

The proposed fault diagnosis method designs the deviating distance of two capacitor voltages to locate open-circuit faults. The deviating distance can accumulate the charging characteristics of the fault sub-module capacitor, and its computation cost is much lower than artificial networks [7,15–17], even lower than the correlation coefficient [11]. Therefore, the fault diagnosis method can detect the fault sub-module before the capacitor voltage is charged very high. After the faulty sub-module is removed, the unequal number of sub-modules that can be put into the arms is the cause of the asymmetrical operation of the

faulty phase. At this time, odd-order circulating currents are introduced into the circulation, of which the fundamental-frequency current is the main component.

As a software-based method, the proposed method has distinct advantages: it can be implemented without additional hardware circuits or sensors, which can greatly improve the modularity of MMC; the fault diagnosis method is based on the analysis of measured capacitor voltage data and is not dependent on the MMC mathematical model, so it is not sensitive to the uncertainty of system parameters and has strong robustness; it has the simplicity of low data-volume and little computation cost; and open-circuit faults can be detected quickly and accurately within a fundamental cycle of 20 milliseconds. Furthermore, the proposed ride-through strategy can decrease the total harmonic distortion (THD), reduce the internal power loss of MMC, and improve the power quality of MMC output, thereby improving the economy and reliability of system operation. The proposed method also could be a supplementary scheme of sub-module capacitance reduction for MMC.

**Author Contributions:** Conceptualization, H.S. and Y.J.; methodology, Y.J.; software, Y.J.; validation, Y.J.; formal analysis, H.S. and Y.J.; investigation, H.S. and Y.J.; resources, Y.J. and M.L.; data curation, Y.J. and M.L.; writing—original draft preparation, Y.J. and M.L.; writing—review and editing, H.S., Y.J. and M.L.; visualization, Y.J. and M.L.; supervision, Y.J. and M.L.; project administration, Y.J. and M.L.; funding acquisition, H.S. All authors have read and agreed to the published version of the manuscript.

**Funding:** This research was funded by the Key Program of the National Natural Science Foundation of China, grant number 52037003, and the Applied Basic Research Key Project of Yunnan Province, grant number 202002AF080001.

**Institutional Review Board Statement:** Not applicable.

**Informed Consent Statement:** Not applicable.

**Data Availability Statement:** Not applicable.

**Conflicts of Interest:** The authors declare no conflict of interest. The funders had no role in the design of the study; in the collection, analyses, or interpretation of data; in the writing of the manuscript; or in the decision to publish the results.

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
