# Peer review of "Fault-Tolerant Control Strategy for Sub-Modules Open-Circuit Fault of Modular Multilevel Converter"

_electronics, doi:10.3390/electronics12051080_

Round 1

Reviewer 1 Report

Authors described Fault-tolerant Control Strategy and properly described theoretical background. Also regarding simulation analysis there are few cases simulated and explained. Authors focused on the case of a single sub‐module open‐circuit fault and only this fault type is analysed. To be correct authors should stated in the title of this work that here is focus on simulation and single module issues and not strategy for the whole range of fault cases. Second issue is that authors present only simulation results and without experimental verification this is not enough to make publication. Experimental verification could be done on low voltage model or on partial assembled low voltage model. In present form article is not acceptable for publishing.

Reviewer 2 Report

Dear authors, while the paper is excellent in shape, there are a few comments and suggestions to improve the manuscript. There is some interest in this type of research, but I found this paper only mildly interesting in its present form. Please strongly consider the following suggestions:

  1. The significance of the study is not clear to me. In this context, please clarify the advantages of this paper in the introduction section because, in the literature, many recent papers consider the same proposed approach. I strongly recommend the authors reconsider the related work section for the literature review and discuss the drawbacks of existing works.
  2. In addition, please emphasize why only the Chinese authors are cited? The subject is treated only by them, or the state of the art has serious flaws, any additional references are missing, and the research is not correctly conducted. I think it's a significant and serious problem. Please carefully review this aspect.
  3. Also, please describe the necessity of each reference in the first section, because multiple citations contradict the ethics of journals with international visibility. Please also reconsider [3-12], [3-5], [6-15], etc.
  4. Section two must be more specific because the methods itself is not well described. There are a lot of equations, but their use in the proposed technique is not precisely defined.
  5. The results analysis is not enough (one page). I cannot see deep analysis related to them and I don't understand the meaning of the results. Please add more analysis. The readers would benefit from a more insightful discussion of the results and a clear statement about the main conclusions drawn from the research carried out.
  6. Why the paper is original please reconsider the Results section with a comparison between the proposed approach and at least one proposed reference (i.e. ref. [27] which is the same).
  7. In the conclusion, the section starts with a brief explanation of the paper's goal and explains what the significant findings are and why your article is essential because the results sections don’t emphasize this aspect.
  8. In this form the paper must be rejected. But, the authors must reconsider the proposed suggestions and the paper con be improvment.

Round 2

Reviewer 1 Report

.

Reviewer 2 Report

In this form the paper can be accepted,